# An improved water correction function for Picarro greenhouse gas analyzers

Friedemann Reum<sup>1</sup>, Christoph Gerbig<sup>1</sup>, Jost V. Lavric<sup>1</sup>, Chris W. Rella<sup>2</sup> and Mathias Göckede<sup>1</sup>

<sup>1</sup>Max Planck Institute for Biogeochemistry, Jena, Germany

<sup>2</sup>Picarro Inc., Santa Clara, CA, USA *Correspondence to*: Friedemann Reum (freum@bgc-jena.mpg.de)

Abstract. Measurements of dry air mole fractions of atmospheric greenhouse gases are widely used in inverse models of atmospheric tracer transport to quantify the sources and sinks of the gases. The measurements have to be calibrated to a common scale to avoid bias in the inferred fluxes. The World Meteorological Organization (WMO) has set requirements for

- the inter-laboratory compatibility of atmospheric greenhouse gas measurements to  $\pm 0.1$  ppm for CO<sub>2</sub> (Southern hemisphere  $\pm 0.05$  ppm) and to  $\pm 2$  ppb for CH<sub>4</sub>. An established series of devices for measurements of greenhouse gas (GHG) mole fractions are the trace gas analyzers manufactured by Picarro, Inc. These have been shown to deliver dry air mole fractions with accuracies within the WMO goals when trace gas signals are measured in wet air and the effects of water vapor are corrected for. Here, we report for the first time on sensitivity of the pressure inside the measurement cavity of Picarro GHG
- analyzers to water vapor. This sensitivity induces biases in the inferred dry air mole fractions of  $CO_2$  and  $CH_4$  if they are obtained using the traditional water correction function. To correct for the pressure effect, we add a pressure-related term to the traditional water correction function, and consider differences between the traditional and enhanced water correction function to be biases of the traditional model. The effect primarily affects low water vapor mole fractions from about 0.05 to about 0.5 %, a domain that has gone undersampled in previous studies of the water correction for Picarro GHG analyzers.
- We observed biases up to about 40 % of the WMO tolerances (80 % for CO<sub>2</sub> in the southern hemisphere). The magnitude of the effect varied across instruments and appeared to be negligible for some, and our experimental results were more robust for CH<sub>4</sub> than for CO<sub>2</sub>. Thus, correction coefficients should be determined for each analyzer individually. Applying our enhanced water correction function improves the accuracy of measurements of dry air mole fractions of CO<sub>2</sub> and CH<sub>4</sub> in humid air with Picarro GHG analyzers on a scale important for keeping the measurement accuracy within the WMO 25 requirements.

## 1 Introduction

Measurements of atmospheric GHG mole fractions are integral data for quantifying the sources and sinks of the gases using inverse models of atmospheric transport (e.g. Kirschke et al., 2013; McGuire et al., 2012). Inverse models require atmospheric measurements calibrated to a common scale, because relative biases in the mole fraction data lead to biases in

the inferred fluxes. The World Meteorological Organization (WMO) has set compatibility goals for atmospheric CO<sub>2</sub> and CH<sub>4</sub> measurements to  $\pm 0.1$  ppm for CO<sub>2</sub> ( $\pm 0.05$  ppm in the southern hemisphere) and  $\pm 2$  ppb for CH<sub>4</sub> (WMO, 2016).

In models of atmospheric greenhouse gas transport, the relevant atmospheric information is the dry air mole fraction, i.e. the number of molecules of the target gas divided by the number of air molecules, not including water vapor. Water vapor is

- excluded because its large variability would cover any signal in the trace gases. To measure dry air mole fractions in the humid atmosphere, there are two strategies: (i) drying the air before measurement and (ii) measuring the wet air signal and correcting for the effects of water vapor later. There are a variety of techniques to dry sample air, including cooling or streaming it through a nafion membrane dryer. Limitations of this strategy include maintenance requirements, minimum limits to the achievable dryness, and possibly effects of drying on the GHG mole fractions (Rella et al., 2013).
- GHG analyzers manufactured by Picarro Inc. (Santa Clara, CA), which are based on the cavity ring-down spectroscopy technique (Crosson, 2008), are used at many sites of the international GHG monitoring network because of their signal stability. When using these instruments, the practice is often not to dry the sample air, but instead to measure wet air mole fractions and subsequently correct for the effects of water vapor. Cavity ring-down spectroscopy is based on absorption of laser light by the target gas inside a resonator cavity. Active temperature and pressure control in the cavity are built-in to
- establish the stable measurement conditions necessary for fitting absorption line shapes. To obtain dry air mole fractions, a water correction function is applied to  $CO_2$  and  $CH_4$  signals measured in wet air. The established form of the water correction function accounts for the dilution effect of water vapor and its effects on the shapes of the absorption lines (Chen et al., 2010; Rella et al., 2013). Dilution changes the target gas mole fraction linearly, and the effects of line shape changes on the mole fraction measurements are modeled as a second-degree Taylor series. Thus, the overall shape of the traditional
- water correction function is a parabola (with a dominant linear term and a small quadratic correction), by which measured wet air mole fractions are divided to obtain dry air mole fractions. By using these functions, water vapor effects on CO<sub>2</sub> and CH<sub>4</sub> can be corrected with residuals below the WMO goals. Better accuracy is achieved by determining individual coefficients for the water correction function specific to each analyzer (Chen et al., 2010; Rella et al., 2013). Here, we report systematic biases of dry air mole fractions inferred using the traditional parabolic water correction functions
- that, to our knowledge, have not been investigated yet. In some cases, the observed systematic deviations were as high as about 40 % of the WMO tolerances (80 % for CO<sub>2</sub> in the southern hemisphere). The largest deviations were at low H<sub>2</sub>O mole fractions around from about 0.05 to about 0.5 %. We found that only few measurement points at such low water vapor mole fractions were sampled in previous studies (Chen et al., 2010; Nara et al., 2012; Rella et al., 2013; Winderlich et al., 2010). In this study, we address the systematic deviations introduced by the traditional water correction function, and find that they
- can be explained by a dependency of the pressure inside the measurement cavity on the water vapor content in the sample air. We present a method to correct for the pressure effect, and quantify the impact this correction has on observations from the field.

#### 2 Methods

## 2.1 Experiments

To determine the effect of water vapor on  $CO_2$  and  $CH_4$  measurements, dry air from pressurized gas tanks was humidified and measured with a Picarro GHG analyzer. Pressure in the measurement cavity of the analyzer was monitored with both the built-in internal pressure sensor and an additional external pressure sensor (Fig. 1).

- To humidify the air stream, two different methods were used. The first approach was designed to allow stable maintenance of defined levels of the water vapor content. The dry air stream was split into two lines, one of which remained untreated. Air in the other line was directed through a gas washing bottle (glass) containing deionized water (depending on bottle size, about 15 ml to about 500 ml were used in the experiments presented here). Thus, air in this line was saturated with water
- vapor (mole fraction about 3 %). Subsequently, the two lines were joined again. The water vapor mole fraction in the rejoined line was controlled by adjusting the flow through the wet and dry lines using needle valves. For one of the experiments (Picarro #1, see Table 1), instead of using the gas washing bottle approach, air was humidified by mixing air from the gas tank and ambient laboratory air. From this experiment, only pressure data were analyzed. The second humidification approach was the droplet method. For these experiments, the humidification unit described above
- was replaced with a tee piece. To humidify the air, a droplet of deionized water (~ 1 ml) was injected into the line through the tee piece using a syringe. Gradual evaporation of this water droplet then caused a gradient over time from high to low water vapor levels in the sample air.

Pressure inside the measurement cavity of Picarro GHG analyzers is kept stable by a feedback loop between a pressure sensor (General Electric NPC-1210) that is mounted inside the cavity, and the outlet valve of the cavity (inlet valve in so-

- called Flight-ready Picarro GHG analyzers, which are customized for airborne measurements). With this loop, the cavity pressure is kept stable at 140 Torr. Since Picarro reports cavity pressure in Torr, we will use this unit throughout this paper (1 Torr = 133.3224 Pa). In our experiments, pressure in the measurement cavity was monitored with an external pressure sensor (General Electric Druck DPI 142) as well as with the internal sensor. To shield the external sensor from humidity changes, it was installed in a dead end branched from the main line behind a drying cartridge filled with magnesium
- perchlorate. The pressure measurement line was branched directly upstream of the Picarro GHG analyzer (downstream for Flight-ready analyzers). To match the pressure inside the cavity to within a few Torr, a needle valve was installed as a choke upstream of the pressure measurement branch (downstream for Flight-ready analyzers). This setup allowed us to monitor pressure independently of water vapor content, while the internal pressure sensor still reacted to changes in water vapor levels in the sampling air.
- The external pressure readings drifted on a timescale relevant for the gas washing bottle experiments. Therefore, in these experiments, dry air was measured between different water vapor levels to calibrate the external pressure readings. Each water vapor level (including dry air) was probed between 15 and 150 minutes (median about 40 minutes) depending on the stability of the external pressure measurement and trace gas readings. For further analysis, average readings from the Picarro

GHG analyzer and the external pressure sensor of the last 10 minutes of each probing interval were used. Some measurements with low water vapor levels with probing times of only about five minutes from an early experiment (Picarro GHG analyzer #1, see Table 1) were included as well. The order of water vapor levels was altered between experiments, including varying from wet to dry, dry to wet, and random switches between dryer and wetter air.

Experiments were performed with five Picarro GHG analyzers (Table 1).

## 2.2 Traditional parabolic water correction function

The effect of water on trace gas measurements made using Picarro GHG analyzers can be described by a water correction function  $f_c(h)$ , where *c* denotes the target gas (here: CO<sub>2</sub> or CH<sub>4</sub>) and *h* is the water vapor mole fraction (measured by the Picarro analyzer). The analyzer measures the wet air mole fraction  $c_{wet}(h)$ , which is related to the dry air mole fraction  $c_{dry}$ 

by the water correction function:

$$c_{dry} = \frac{c_{wet}(h)}{f_c(h)} \tag{1}$$

The traditional form of the water correction function takes into account dilution and line shape effects (details in Sect. 1). These are described by a second-degree Taylor series, i.e. a parabola:

 $f_c(h) = 1 + a_c \cdot h + b_c \cdot h^2$ 

(2)

The coefficients  $a_c$  and  $b_c$  can be derived from water correction experiments such as those described in Sect. 2.1.

## 3 Results

### 15 3.1 Links among external pressure measurement, CO<sub>2</sub>, CH<sub>4</sub>, H<sub>2</sub>O and cavity pressure

To establish the link among the external pressure measurement,  $CO_2$ ,  $CH_4$ , and internal cavity pressure for each Picarro GHG analyzer, the cavity pressure was varied manually using Picarro Inc. software in the range observed when varying water vapor content. Externally measured pressure,  $CO_2$ , and  $CH_4$  varied linearly with internally monitored cavity pressure, with similar slopes for all instruments. As an example, the values for Picarro #3 obtained in dry air are shown in Table 2.

The water vapor measurement was not sensitive to cavity pressure. The slopes in wet air (3 % H<sub>2</sub>O) were measured for Picarro #3 and were very similar to the slopes in dry air (CO<sub>2</sub>: +5 %, CH<sub>4</sub>: -2 %, cavity pressure: +1 %). Hence, internal cavity pressure, CO<sub>2</sub> and CH<sub>4</sub> were modeled as linear functions of externally measured pressure for subsequent analyses.

#### 3.2 Dependency of cavity pressure on water vapor content

We monitored cavity pressure using an external sensor during gas washing bottle experiments (Sect. 2.1) for three different Picarro GHG analyzers (Table 1). The pressure readings of the internal pressure sensors were, as expected, stable at 140 Torr with standard deviations of 0.015 Torr or less. To calculate a "corrected cavity pressure" from the external pressure

(4)

measurement, pressure readings for dry air before and after each wet air measurement were interpolated to the times of the wet air measurements. The deviations between the wet air pressure values and the interpolated dry air pressure values were multiplied with the slope described in Sect. 3.1, and added to the dry air cavity pressure of 140 Torr. The corrected cavity pressure obtained in this way varied systematically with the water vapor mole fraction of the sample air (Fig. 2). The

5 variations displayed a uniform pattern for all three instruments. The pressure dropped in the presence of water vapor, and the gradient of pressure with respect to water vapor was larger between 0 and about 0.2 % H<sub>2</sub>O than for higher water vapor content, exhibiting a "bend" where the two regimes meet. The deviations were up to 0.5 Torr for 3 % H<sub>2</sub>O. We describe the relationship between cavity pressure and water vapor mole fraction with an empirical function:

$$p(h) = p_0 + s \cdot h + d_p \cdot \left(e^{-\frac{h}{h_p}} - 1\right)$$
(3)

In this equation, p is the cavity pressure as determined from the external pressure measurement, h is the water vapor mole fraction,  $h_p$  is the position of the pressure bend described above,  $d_p$  is a measure for the pressure gradient at  $0 \% < h < h_p$ ,  $p_0$  is the pressure in dry air (fixed at 140 Torr), and s is the slope for  $h \gg h_p$ . Note that this empirical fit function is valid only in the water vapor range covered by measurements (see Fig. 2). Data from droplet experiments suggest that the pressure

variation does not continue linearly along the slopes derived here at higher water vapor levels, so an extrapolation is not recommended.

All free parameters of the pressure model  $(s, d_p \text{ and } h_p)$  varied between instruments (Table 3). For the empirical water correction functions for CO<sub>2</sub> and CH<sub>4</sub>, only the pressure bend position  $(h_p)$  is relevant, as will be shown later. The mean estimate of  $h_p$  from all three experiments was (mean and standard deviation):

$$h_p = (0.079 \pm 0.014) \% H_2 O$$

## 3.3 Correction of the pressure effect on CO<sub>2</sub> and CH<sub>4</sub>

Reliable data for both pressure and the target gases  $CO_2$  and  $CH_4$  were obtained in one experiment (with Picarro #3), which is presented in this section. Based on the data, four models were tested as potential water correction functions for  $CO_2$  and  $CH_4$  to examine performance, robustness, transferability, and consistency of the results. Model (i) was the traditional

parabolic function, Eq. (2). The other three models represent different strategies to correct for the pressure effect. Model (ii) consisted of first correcting the pressure effect on the wet air mole fractions by estimating the pressure bias from Eq. (3) and then correcting the trace gas bias using the sensitivity of  $CO_2$  and  $CH_4$  to pressure (Table 2); then the traditional parabolic water correction was applied to the corrected wet air mole fractions. For models (iii) and (iv), the traditional parabolic model was expanded to account for the pressure effect. Since the trace gas readings of the analyzer varied linearly with pressure

(Sect. 3.1), the pressure effect was described as in Eq. (2), i.e. as a linear and a non-linear term. Since a linear dependency is already accounted for in the traditional parabolic model, the non-linear part of Eq. (2) was added to Eq. (3) to obtain a new model for the water correction:

$$f_c^p(h) = \underbrace{\mathbf{1} + \mathbf{a}_c \cdot \mathbf{h} + \mathbf{b}_c \cdot \mathbf{h}^2}_{f_c(h)} + \mathbf{d}_c \cdot \left(e^{-\frac{h}{h_p}} - \mathbf{1}\right)$$
(5)

The parameter  $d_c$  describes the magnitude of the pressure change at low water vapor contents and sensitivity of the target

- 5 gas to the pressure change. The parameter  $h_p$  is the pressure bend position from Eq. (3). In model (iii), the parameters  $a_c$ ,  $b_c$ ,  $d_c$  and  $h_p$  from Eq. (5) were fitted to the trace gas data. Model (iv) was the same as model (iii) except that the pressure bend position  $h_p$  was set to the value obtained from the pressure data. Since all free parameters in model (iii) were estimated from the available trace gas data, this model was the most consistent with the data. Therefore, in subsequent analyses, we assumed that the fit to model (iii) yielded the true water correction function. Hence, we used the differences between the results from 10 this model and the others as estimates of their biases.
- The experiment was performed with dry air mole fractions of 404.0 ppm  $CO_2$  and 1842 ppb  $CH_4$ . Water-corrected  $CO_2$  and  $CH_4$  data from this experiment are shown in Fig. 3. For  $CH_4$ , the most striking visible feature was the wave-like structure in the dry air mole fractions when using model (i), the traditional parabolic water correction function. The maximum negative bias of this model was 0.85 ppb at 0.17 %  $H_2O$  (corresponding to 0.046 % of the dry air mole fraction), and the maximum
- 15 positive bias was 0.37 ppb at 1.7 % H<sub>2</sub>O. Hence, the peak-to-peak difference was 1.2 ppb. The standard deviation of the dry air mole fractions estimated with this model was 0.35 ppb. By contrast, no structure was visible in the dry air mole fractions calculated with any of the three formulations taking into account the pressure change. This is reflected in the lower standard deviations of the dry air mole fractions, which were 0.20 ppb for model (ii), 0.17 ppb for model (iii) and 0.18 ppb for model (iv). This demonstrates the improvement achieved by correcting for the effect of pressure bias on CH<sub>4</sub>.
- 20 The result from model (ii) yielded slightly larger deviations from the mean than models (iii) and (iv) in the range 0.1-0.3 % H<sub>2</sub>O. These deviations were compatible with a sensitivity of CH<sub>4</sub> to cavity pressure changes of 80 % of the value inferred in Sect. 3.1, since at this value the results from model (ii) resemble the results from model (iv). This discrepancy is discussed in Sect. 4.3.

The pressure bend position estimated from the CH<sub>4</sub> data was  $h_p = (0.059 \pm 0.015) \%$  H<sub>2</sub>O. This is smaller than the estimate based on pressure data of  $h_p = (0.095 \pm 0.011) \%$  H<sub>2</sub>O. Despite this discrepancy, the two models yielded very similar dry air mole fractions, with differences within 0.12 ppb and a peak-to-peak difference of 0.22 ppb between 0.05 and

0.39 % H<sub>2</sub>O. This demonstrates the robustness of the method against uncertainties in  $h_p$ .

The wave-like structure seen in the CH<sub>4</sub> dry air mole fractions estimated using the traditional parabolic water correction function was absent in the CO<sub>2</sub> data for this instrument. The standard deviations of the dry air mole fractions were similar for all models (model (i): 0.017 ppm, model (ii): 0.021 ppm, models (iii) and (iv): 0.014 ppm).

Using model (ii) induced a bias similar to the one present in the results of model (i) for  $CH_4$  but with opposite sign (Fig. 3). This hints at an overcompensation of the pressure effect on  $CO_2$ . Indeed, following the same argument as for  $CH_4$ , the results of model (iv) were reproduced when the sensitivity of  $CO_2$  to cavity pressure changes was set to 35 % of the value presented in Table 2. Note that the wave-like structure in  $CO_2$  was apparent for one other instrument (Picarro #5, see Sect. 3.4 and 3.5.1). This has not been presented in this section since no external pressure sensor data were obtained for this instrument.

## 3.4 Consistency across instruments

To investigate whether common coefficients applicable to all Picarro GHG analyzers can be given for the enhanced water correction function, we performed water correction experiments with several Picarro GHG analyzers. Although reliable pressure and trace gas data from a single experiment were obtained for only one instrument (presented in Sect. 3.3), trace gas

- data from water correction experiments were obtained for two more analyzers. The pressure effect on the trace gas data from these instruments differed in magnitude. Out of the three instruments for which trace gas data were investigated (Picarros #3, #4, and #5), two exhibited the pressure effect visibly for CH<sub>4</sub> (Fig. 3 and Fig. 5; the exception was Picarro #4, see Fig. 4). In contrast, the CO<sub>2</sub> measurements of two instruments (#3 and #4) seemed to be unaffected by pressure changes (Fig. 3 and Fig. 4; the exception was Picarro #5, see Fig. 5). The differences make clear that common coefficients applicable to all Picarro
- GHG analyzers can not be given based on our data. This is further discussed in Sect. 4.3 and Sect. 4.4.

## 3.5 Application to ambient measurements

In this section, we demonstrate the impact that the pressure effect has on ambient GHG observations from a site in northeast Siberia. The site is located in the remote village of Ambarchik on the coast of the Arctic Ocean (69.62° N, 162.30° E), and has been operational since August 2014. Air inlets are at 27 m and 14 m above ground level, and probed in turns for intervals

of 15 and 5 minutes, respectively. The gases CO<sub>2</sub> and CH<sub>4</sub> are measured in the humid air stream with a Picarro G2301 analyzer (Picarro #5). The measurements are calibrated automatically by measuring gas tanks calibrated to the WMO scales X2007 (CO<sub>2</sub>) and X2004A (CH<sub>4</sub>) every 116 hours.

#### 3.5.1 Deriving coefficients for the improved water correction function without pressure data

- In this section, we derive coefficients for the improved water correction function, Eq. (5), for CO<sub>2</sub> and CH<sub>4</sub> for the Picarro
  G2301 analyzer operated in Ambarchik. For this instrument, one water correction experiment with the gas washing bottle method (see Sect. 2.1) was performed, using dry air from a pressurized tank with mole fractions of 352.9 ppm CO<sub>2</sub> and 1797 ppb CH<sub>4</sub>. Cavity pressure was not monitored during this experiment. We estimated the parameters a<sub>c</sub>, b<sub>c</sub> and d<sub>c</sub> from the trace gas data (Table 4 and Table 5), but the number of data points was insufficient to also estimate the pressure bend position h<sub>p</sub>. Therefore, we used the mean h<sub>p</sub> from the three experiments with external pressure monitoring, Eq. (4), and
  investigated the uncertainty associated with this procedure.
  - 7

As a conservative estimate, we considered an interval of three standard deviations around the mean a plausible range for  $h_p$ , i.e.  $h_p \in [0.036, 0.122] \ \text{M}_2\text{O}$ . Varying  $h_p$  in small steps within this interval, we fitted the other parameters of Eq. (5) to the trace gas data. To assess the uncertainty associated to using the mean  $h_p$ , we assumed that one of the  $h_p$  yielded the true water correction function for this instrument, and determined whether using the mean  $h_p$  could induce a larger error than

- using the traditional parabolic correction. For this assessment, we compared not only the data points obtained during the experiment, but sampled the fitted functions at 999 evenly spaced points over the range of H<sub>2</sub>O covered during the experiment. For CO<sub>2</sub>, the maximum deviation between the function using the mean  $h_p$  and any other  $h_p$  in the plausible range was 0.007 ppm, while the best result for the maximum deviation between the traditional parabolic correction and the improved functions with  $h_p$  in the plausible range was 0.017 ppm. For CH<sub>4</sub>, the deviations were 0.20 ppb and 0.35 ppb,
- respectively. Therefore, maximum bias due to the uncertainty of  $h_p$  was smaller than the minimum bias when using the traditional parabolic correction for both CO<sub>2</sub> and CH<sub>4</sub>.

For reference, we also investigated the values for  $h_p$  estimated from the trace gas data. Fitting Eq. (5) to CH<sub>4</sub> yielded  $h_p = (0.086 \pm 0.053) \%$  H<sub>2</sub>O, close to the mean  $h_p$  from the three pressure experiments but with a large uncertainty. For CO<sub>2</sub>, the fit yielded  $h_p = (0.34 \pm 0.19) \%$  H<sub>2</sub>O, outside of the range considered plausible for  $h_p$  and again with a very large

- uncertainty. It may be argued that the plausible range  $h_p$  should be extended to 0.34 % H<sub>2</sub>O. Using this extended range, the argument for the benefit of the improved water correction over the traditional one despite the uncertainty of  $h_p$  was weaker, but still held. However, we argue that the estimate  $h_p = 0.34$  % H<sub>2</sub>O is probably far from the true pressure bend position for this instrument. First, the estimate was based on CO<sub>2</sub> data, which were less consistent with pressure data than CH<sub>4</sub> data for another instrument (Sect. 3.3). Second, it is far from the pressure bend positions of all three instruments for which pressure
- data were obtained (Sect. 3.2). Third, the estimate was based on very few data points and was not robust against jackknife resampling (not shown). For these reasons, we consider  $h_p$  from Eq. (4) to be a realistic estimate for this Picarro analyzer. To assess the bias of the traditional parabolic fit function, we assumed that the fit with pressure correction term using the mean  $h_p$  was the true water correction function. Maximum absolute deviations between the two were 0.037 ppm CO<sub>2</sub> and 0.78 ppb CH<sub>4</sub> at H<sub>2</sub>O = 0.16 %, and peak-to-peak deviations were 0.052 ppm CO<sub>2</sub> and 1.11 ppb CH<sub>4</sub> between H<sub>2</sub>O = 0.18 %
- and 1.8 %. These deviations corresponded to biases of 0.0104 % CO<sub>2</sub> and 0.0437 % CH<sub>4</sub>. As an example, for dry air mole fractions of 400 ppm CO<sub>2</sub> and 2000 ppb CH<sub>4</sub>, the maximum absolute bias of the traditional water correction function would be 0.042 ppm CO<sub>2</sub> and 0.87 ppb CH<sub>4</sub>.

In Sect. 3.3, we compared the sensitivities of  $CO_2$  and  $CH_4$  to cavity pressure inferred from controlled cavity pressure changes with those inferred from water vapor changes for Picarro #3. We made the same comparison for Picarro #5 under

30 the assumptions that the cavity pressure dependency on water vapor and the sensitivities of the trace gases to controlled cavity pressure changes were the same as for Picarro #3. The data from the water correction experiment were compatible with a sensitivity of CO<sub>2</sub> to cavity pressure change of 60 % of the value from controlled cavity pressure changes, which was