# Peer review of "An improved water correction function for Picarro greenhouse gas analyzers"

_Atmospheric Measurement Techniques, 2017_

## Referee Comment (RC1) · Anonymous Referee #1 · 6 Jul 2017

**General comments**

This paper describes the high accuracy correction function for the water vapor interferences on the CO2 and CH4 measurements using the wavelength-scanned cavity ring-down spectroscopy, which has been widely used instrument for the atmospheric greenhouse gas measurements. While several past studies proposed empirical correction function for the water vapor interferences expressed as second order polynomial model, the authors pointed out the presence of additional water vapor interference on the cavity pressure measurements, which results in undercorrection of the CO2 and CH4 measurements using the empirical correction function for humid air samples, especially at low water vapor content. Based on their experiments, the authors improved the empirical correction function and demonstrated that significant differences

occurred in the correction values between the previous and the improved correction function when these functions were applied to the field observation data.

The paper's topic is interesting, and may be an important contribution for the atmospheric greenhouse gas measurement community. However, I feel that the manuscript unfortunately suffers from redundancy, unclear writing, bad organization, and confusing data analysis. All these problems make it extremely difficult to follow. Furthermore, significance of the measurement biases due to the water vapor interference on the cavity pressure measurements was inconsistent across the instruments. I am uncertain of this study, and therefore I think it would be better to revise the experimental methodology carefully and needs further investigation. I am very afraid, but I suggest rejection of this manuscript from AMT. I encourage the author to rewrite the manuscript from scratch with the help of the coauthors for clarity after consideration of my comments.

Specific comments (Major comments)

I have great concern about the experiment for the estimation of the quantitative relationship among the readings of external pressure sensor, CO2, CH4, and internal pressure sensor. The author used the Mg(ClO4)2 cartridge to shield the external pressure sensor from humidity change. The external pressure sensor measurements can also be biased by the presence of water vapor? Then I wonder why the author did not use the pressure sensor independent of the water vapor presence for the experiments. Since the experimental system can be highly complex due to the installation of the Mg(ClO4)2 cartridge, I have no idea what the external pressure sensor measures. In addition, there are several other concerns as described below:

1. What was temperature control for the humidification unit? The slight temperature change will affect the solubility of CO2 and CH4 in the de-ionized water which results in change in the mole fractions of CO2 and CH4 in the sample air, especially for CO2.

2. There is no detailed information for the Mg(ClO4)2 reagent, but the author used CO2-saturated Mg(ClO4)2 reagent to avoid CO2 loss on the reagent?
3. The author used the needle valve to adjust the pressure readings close to those of the internal pressure sensor, but what was the stability of the sample pressure downstream the needle valve? The pressure change can cause the increased CO2 absorption/desorption on Mg(ClO4)2 reagent.

4. Depending on the water vapor absorption on the Mg(CIO4)2 reagent, magnitude of the pressure loss in the Mg(CIO4)2 cartridge may be changed, resulting in the pressure gradient between up- and downstream the cartridge.

5. The author checked complete removal of water vapor behind the Mg(ClO4)2 cartridge at the external pressure sensor?

---

## Author Comment (AC1) · 7 Jul 2017

Among some other points, the main concerns of the reviewer are (1) the influence of the drying cartridge, which was used for shielding the external pressure sensor against water vapor changes, especially on CO2, and (2) the validity of the relationship between external pressure sensor reading and Picarro cavity pressure, due to certain components of the experimental setup (drying cartridge, needle valves). As we explain in our attached reply, concern (1) is unfounded due to our experimental setup. Concern (2) requires more attention and we acknowledge that there are uncertainties in the external pressure readings (however, these are discussed in the manuscript). We address all comments of the reviewer in the attached reply. After reading the review, we believe that in our efforts to write a concise paper we may have kept certain

sections too brief. In a revised manuscript, we will add the clarifications given in the responses in the cases where they were not present in the manuscript that we initially submitted. However, we would like to emphasize at this point that the uncertainties regarding the external pressure measurement have no influence on the main message of our study, i.e. improving the empirical water vapor correction for CO2 and CH4 readings of Picarro GHG analyzers. One of our major results was that coefficients for the improved empirical water correction can be obtained even without external pressure measurements. The external pressure measurements were mainly used to infer whether or not the observed shortcomings of the traditional water vapor correction – i.e. systematic, water-dependent biases in the corrected CO2 and CH4 data – were artifacts of water correction experiments and should thus be ignored. Even though there are uncertainties associated with the use of the external pressure sensor, the information obtained from this instrument served very well for this specific purpose: Our experiments with external pressure monitoring revealed that the shortcomings of the traditional water vapor correction can be linked to pressure changes in the cavity of the Picarro analyzer, and therefore should be corrected for. Accordingly, as the main objective of the presented study we provided a way to correct for the effect. This summary statement and the more detailed comments in the attached response aim at clarifying the rather minor role of the accuracy of the external pressure measurements. We believe that the concerns raised by the reviewer regarding this element of our study should not put the validity of the overall findings into question.

Please also note the supplement to this comment:
https://www.atmos-meas-tech-discuss.net/amt-2017-174/amt-2017-174-AC1-supplement.pdf

[Figure]

**Supplement:**

Friedemann Reum, Christoph Gerbig, Jost V. Lavric, Chris W. Rella and Mathias Göckede

**Summary:**

Among some other points, the main concerns of the reviewer are (1) the influence of the drying cartridge, which was used for shielding the external pressure sensor against water vapor changes, especially on CO2, and (2) the validity of the relationship between external pressure sensor reading and Picarro cavity pressure, due to certain components of the experimental setup (drying cartridge, needle valves). As we explain below in replies to the reviewer's specific comments, concern (1) is unfounded due to our experimental setup. Concern (2) requires more attention and we acknowledge that there are uncertainties in the external pressure readings (however, these are discussed in the manuscript). We address all comments of the reviewer below.

After reading the review, we believe that in our efforts to write a concise paper we may have kept certain sections too brief. In a revised manuscript, we will add the clarifications given in the responses below in the cases where they were not present in the manuscript that we initially submitted.

However, we would like to emphasize at this point that the uncertainties regarding the external pressure measurement have no influence on the main message of our study, i.e. improving the empirical water vapor correction for CO2 and CH4 readings of Picarro GHG analyzers. One of our major results was that coefficients for the improved empirical water correction can be obtained even without external pressure measurements. The external pressure measurements were mainly used to infer whether or not the observed shortcomings of the traditional water vapor correction – i.e. systematic, water-dependent biases in the corrected CO2 and CH4 data – were artifacts of water correction experiments and should thus be ignored. Even though there are uncertainties associated with the use of the external pressure sensor, the information obtained from this instrument served very well for this specific purpose: Our experiments with external pressure monitoring revealed that the shortcomings of the traditional water vapor correction can be linked to pressure changes in the cavity of the Picarro analyzer, and therefore should be corrected for. Accordingly, as the main objective of the presented study we provided a way to correct for the effect.

This summary statement and the more detailed comments below aim at clarifying the rather minor role of the accuracy of the external pressure measurements. We believe that the concerns raised by the reviewer regarding this element of our study should not put the validity of the overall findings into question.

**Reviewer's comment:**

The paper's topic is interesting, and may be an important contribution for the atmospheric greenhouse gas measurement community. However, I feel that the manuscript unfortunately suffers from redundancy, unclear writing, bad organization, and confusing data analysis. All these problems make it extremely difficult to follow. Furthermore, significance of the measurement biases due to the water vapor interference on the cavity pressure measurements was

inconsistent across the instruments. I am uncertain of this study, and therefore I think it would be better to revise the experimental methodology carefully and needs further investigation. I am very afraid, but I suggest rejection of this manuscript from AMT. I encourage the author to rewrite the manuscript from scratch with the help of the coauthors for clarity after consideration of my comments.

**Author's response:**

The reviewer made a general statement about "redundancy, unclear writing, bad organization, and confusing data analysis" of/in the manuscript, and suggested "to rewrite the manuscript from scratch with the help of the coauthors for clarity after consideration of [his/her] comments". We would hereby like to emphasize that before submission the manuscript went through numerous draft iterations, involving input and feedback from all coauthors. All persons listed on the author list fully agreed with the content of the manuscript, and the way it was presented. We acknowledge that it is always possible to improve the presentation of a text by further polishing certain aspects. Still, we find the structure and writing of the manuscript are of adequate quality to deliver the scientific message behind our study. We are open to constructive suggestions how to further improve the presentation, and therefore would like to ask the reviewer for more specific criticism which aspects of our text need revision.

The fact that the observed effect had different magnitudes for the individual instruments was addressed in the manuscript. As described in sections 3.4 and 4.4 of the manuscript, this observation has but one consequence for the main point of our study, which is that coefficients for our improved empirical correction have to be obtained per instrument (as opposed to using common coefficients for all instruments). The underlying reasons for the differences between instruments may be subject to future research.

**Reviewer's comment:**

Specific comments (Major comments)

I have great concern about the experiment for the estimation of the quantitative relationship among the readings of external pressure sensor, $CO_2$, $CH_4$, and internal pressure sensor. The author used the $Mg(ClO_4)_2$ cartridge to shield the external pressure sensor from humidity change. The external pressure sensor measurements can also be biased by the presence of water vapor? Then I wonder why the author did not use the pressure sensor independent of the water vapor presence for the experiments. Since the experimental system can be highly complex due to the installation of the $Mg(ClO_4)_2$ cartridge, I have no idea what the external pressure sensor measures. In addition, there are several other concerns as described below:

**Author's response:**

The reviewer raises the question whether the relationships between the readings of external pressure sensor, $CO_2$, $CH_4$, and internal pressure sensor were valid over the course of a whole experiment, focusing on the validity of the external pressure measurement. These questions were addressed in sections 3.1, 4.1.2 and 4.3 of the manuscript. We kept especially sections 3.1 and 4.1.2 short in order to focus on the major results of the study, and will provide more details here in a revised manuscript version. The relationships the reviewer refers to may change slightly with time and/or water content. Still, as presented in section 3.1 of the manuscript, the relationships were not considerably affected by water vapor in the measured air, and moreover were very similar across instruments. Regarding changes of the relationships with time (e.g. due to

saturation of the drying agent with water) we randomized the order of water levels probed in one experiment (Picarro #3, mentioned in sect. 2.1 of the manuscript). With this setup, a drift of the external pressure sensor would have been visible in random biases of the pressure versus water level. This was plotted in Fig. 2 of the manuscript, and the residuals to the fit were small compared to the observed effect. Therefore, if a drift such as the one suggested by the reviewer was present, it was smaller than the signal. To illustrate the different probing strategies, we present the order in which water levels were probed for all instruments with external pressure measurement in the figure below.

[Figure]

[Figure]

Fig. 1: Water levels probed during the experiments with external pressure monitoring.

These considerations do not address whether there was a bias present in the external pressure measurement during wet air measurements. On the one hand, we reported that the slope of internal cavity pressure versus external pressure measurement was independent of water vapor (sect. 3.1 of the manuscript). On the other hand, we cannot exclude a water-dependent offset based on our data. Given that the shape of the pressure changes – in particular the "pressure bend" – was consistent between pressure and $CH_4$ data, we omitted this in the discussion of the uncertainties of the external pressure measurement. Instead, we concluded this discussion by emphasizing that uncertainties in the external pressure measurement are insignificant for the water correction of $CO_2$ and $CH_4$, since all parameters can and should be derived from the $CO_2$ and $CH_4$ data directly (sect. 4.1.2. in the manuscript). Furthermore, we only briefly discussed such a bias as a potential reason for observed discrepancies between the external pressure data and trace gas data (sect. 4.3. in the manuscript). We explained why this hypothesis is less likely than another one. However, in a revised manuscript, we will include the statement that a water-dependent offset may nonetheless be present, although it is unlikely to explain the discrepancies discussed in sect. 4.3 of the manuscript.

We are not certain about what the reviewer means by "why the author did not use the pressure sensor independent of the water vapor presence for the experiments". We measured the pressure of the air measured by the Picarro analyzer, so the external pressure sensor had to probe the wet air stream. To minimize influence of water vapor on the external pressure sensor, it was installed with the drying cartridge in a dead end branched from the air stream. If there are ways how to circumvent this issue, we are certainly open to more specific suggestions by the reviewer.

**Reviewer's comment:**

1. What was temperature control for the humidification unit? The slight temperature change will affect the solubility of $CO_2$ and $CH_4$ in the de-ionized water which results in change in the mole fractions of $CO_2$ and $CH_4$ in the sample air, especially for $CO_2$.

**Author's response:**

We did not discuss temperature stability in detail in the manuscript for the sake of brevity. Instead, we briefly described the strategies for avoiding the impact of temperature effects and other drifts, and will provide more information below.

As the reviewer stated, the solubilities of $CO_2$ and $CH_4$ vary with temperature. This would lead to outgassing or dissolution of the trace gases in the water reservoir in our experiments, thus changing the mole fractions of the gases in air. However, $CO_2$ and $CH_4$ in the air stream would only be affected until the water reservoir reached equilibrium with the air stream. The equilibration time depends on several factors, among them the efficiency of the mixing of the air stream with the water reservoir, the flow rate of the air stream, the volume of the water reservoir, and in our experiments the head space of the gas washing bottle. In the manuscript, we acknowledged that equilibration effects may have gone unobserved if they occurred on a timescale much longer than an hour (sect. 4.3). However, we employed two strategies to exclude impacts of effects related to equilibration or drift of any kind (temperature, pressure in cavity, external pressure measurement, outgassing and dissolution). One strategy was to wait for stabilization of $CO_2$, $CH_4$ and external pressure readings before using the data to obtain a data point (hence the different probing times per water level, see sect. 2.1 of the manuscript). The other strategy was to vary the order in which different water vapor levels were probed during the

experiments (see sect. 2.1 of the manuscript and Fig. 1 in this response). In particular, one experiment (Picarro #3) was carried out using a random sequence of water vapor levels. It is unlikely that a temperature or any other drift correlated with this sequence (see sect. 4.3 of the manuscript).

We did not discuss temperature stability in the manuscript and will do so in the following paragraph.

The strategy for temperature stability was to keep the temperature in the whole laboratory stable. Reasons for focusing the temperature stabilization not only on the humidification unit were the risk of condensation between the gas washing bottle and the Picarro analyzer, and the possibility of different rates of heat exchange between the air flowing through the gas washing bottle and the water reservoir due to different flow rates over the course of each experiment. Experiments with Picarros #1–#4 were carried out in an air-conditioned laboratory, but unfortunately there is no temperature data available for these experiments. However, we do have data of the room temperature during the experiment with Picarro #5. Temperature stability during this experiment was difficult to achieve, as it took place at a remote site in Siberia, and we managed to keep the room temperature within $\pm$ 0.3 K. Below, we plotted the corrected $CO_2$ and $CH_4$ data and air temperature against time (Fig. 2). A temperature effect should be reflected in the dry air mole fractions of $CO_2$ and $CH_4$. The figure shows that the magnitude of $CO_2$- and $CH_4$ variations was well below the magnitude of the effect we corrected using the improved water correction function (0.037 ppm $CO_2$ and 0.78 ppb $CH_4$, see sect. 3.5.1 of the manuscript). This means that even if a temperature effect was at play, its effect was small compared to the bias corrected by applying our correction.

Despite all our efforts to minimize the effects of temperature on our measurements, we discussed this effect as a potential explanation for the inconsistent results for $CO_2$ in sect. 4.3 in the manuscript, and concluded that, although it does not explain the observations perfectly, is the most likely of the explanations we considered. Related to this is the following statement from sect. 4.4 of our manuscript (page 12, lines 11-13):

"In some cases, the effect on $CO_2$ (two out of three instruments) and $CH_4$ (one of these two instruments) even appeared negligible. In those cases, it may be possible that a small effect exists but is masked by random fluctuations."

To summarize, we already acknowledged the possibility of effects on $CO_2$ related to its high solubility in water (compared to $CH_4$) in sect. 4.3 of the manuscript. To further clarify the potential impacts of this effect on the results obtained within the context of this study, in a revised version of the manuscript we will expand this section and section 4.4 with the remarks on temperature from this response.

[Figure]

Fig. 2: Water vapor, corrected CO2 and CH4 data and Temperature vs time during the experiment with Picarro #5

**Reviewer's comment:**

2. There is no detailed information for the Mg(ClO4)2 reagent, but the author used CO2-saturated Mg(ClO4)2 reagent to avoid CO2 loss on the reagent?

**Author's response:**

Before we address this comment, we would like to make sure there was no misunderstanding about the experimental setup: this and other comments by the reviewer suggest he/she had in mind a setup where the air stream measured using the Picarro analyzer flowed through the drying cartridge. To exclude a misunderstanding, we stress that the cartridge was instead installed in a dead end branch (see also sect. 2.1 and Fig. 1 in the manuscript). More importantly regarding this particular comment in the context of our manuscript, the experimental results that could have

been affected by the drying cartridge – the ones from which we interpreted the CO2 (and CH4) data – were either performed without external pressure measurements, and thus no drying cartridge was installed at all (Picarros #4 and #5), or the cartridge was installed downstream of the Picarro analyzer (Picarro #3). Thus, the cartridge could have no direct effect on CO2 and CH4 mole fraction measurements of the Picarro analyzers in our experiments.

The effect the reviewer pointed out in this comment may become relevant when external pressure readings are obtained using ground-Picarros, since the dead end branch containing the drying cartridge would be upstream of the Picarro analyzer (see Fig. 1 of the manuscript). However, since the drying cartridge would be installed in a dead end branch, even with such a setup we would expect very little influence on CO2 and CH4 mole fractions in the air stream measured by the Picarro analyzer, especially since the distance between the measured air stream and the drying cartridge can be increased by using longer tubing.

Perhaps we did not state clearly enough in the manuscript that some experiments were performed without external pressure measurement. This information was contained in Table 1. We will add a paragraph to section 2.1 about the experiments that were performed without external pressure measurement.

**Reviewer's comment:**

3. The author used the needle valve to adjust the pressure readings close to those of the internal pressure sensor, but what was the stability of the sample pressure downstream the needle valve? The pressure change can cause the increased CO2 absorption/desorption on Mg(ClO4)2 reagent.

**Author's response:**

Regarding part one of this comment: We indeed observed a drift of the external pressure sensor over time (see Fig. 3 below). The needle valve in question may play a role in this context. The drift was only briefly mentioned in the manuscript, alongside the mitigation strategy: we took dry air measurements between wet air measurements and used the pressure differences as measurement points (this was described in section 2.1 of the manuscript). This procedure relies on the assumption that the slope of internal pressure change versus external pressure reading (sect. 3.1) was not significantly affected by the drift of the external pressure reading. This assumption was not verified in a separate experiment, but discussed in section 4.1.2 of the manuscript. In this section, we established that the difference in the slope of the external pressure sensor readings versus internal cavity pressure between dry and wet air measurements was negligible (sect. 3.1). As outlined in the response to first specific comment of the reviewer, this does not exclude a water vapor-dependent offset during wet air measurements. As stated in said response, we will include a statement about this possible offset in a revised manuscript.

An argument for why pressure stability was sufficient with respect to time (not water vapor), which was not contained in the manuscript, was the magnitude of the drift, which was 0.1 Torr during two experiments (Picarro #1 and #3). This was less than the pressure difference between dry and wet air (0.4 Torr). Therefore, we argue that pressure drift with time, which may be attributable to the needle valve, had no considerable influence on the external pressure readings. During the experiment with Picarro #2, the external pressure reading drifted with a larger magnitude, 1.2 Torr. For this instrument, the argument above may not hold, but we observed only

small differences irrelevant to our findings between the pressure readings from this experiment and the ones from the other two.

[Figure]

**Fig. 3: Drift of the external pressure measurement during dry air measurements for all experiments with external pressure measurement.**

Part two of this comment was addressed in the answer to comment #2 of the review (see above): the drying cartridge was never in contact with air before it was measured by the Picarro analyzer for those experiments from which we analyzed the CO2 and CH4 readings, so it could not have had an effect via absorption/desorption.

**Reviewer's comment:**

4. Depending on the water vapor absorption on the Mg(ClO4)2 reagent, magnitude of the pressure loss in the Mg(ClO4)2 cartridge may be changed, resulting in the pressure gradient between up- and downstream the cartridge.

**Author's response:**

We already discussed the influence of the drying cartridge and other factors on the external pressure measurement in detail in the answers to the other comments above, and stated that we will include these statements in a revised version of the manuscript.

This comment may be understood as suggesting the possibility that the drying cartridge caused pressure changes in the cavity of the Picarro analyzer. This is impossible, since the pressure stabilization of the instrument compensates external pressure changes. Thus, an effect of the drying cartridge on CO2 and CH4 via pressure variations can be excluded. To underline this statement, the systematic water-dependent biases of CO2 and CH4 were also present when no drying cartridge was installed (Picarro #5).

**Reviewer's comment:**

5. The author checked complete removal of water vapor behind the Mg(ClO4)2 cartridge at the external pressure sensor?

**Author's response:**

This is another possibility for inaccuracies of the external pressure measurement. We did not check for complete removal of water vapor behind the drying cartridge, and will include this alongside the other considerations raised by the reviewer and discussed in this response as stated above.

We would like to conclude by stating again that uncertainties of the external pressure sensor would not influence CO2 and CH4 readings of the Picarro analyzer, and that external pressure measurements are not necessary to achieve the main objective of this study: a better water correction for CO2 and CH4 for Picarro GHG analyzers.

---

## Referee Comment (RC2) · Anonymous Referee #2 · 2 Aug 2017

This paper focuses on improving the accuracy of GHG dry mole fraction measurements in humid air made by Picarro cavity ring-down spectrometers. The authors derived the sensitivity of the cavity pressure to water vapor, and presented an enhanced water correction function by introducing an additional term to the traditional parabolic water correction function, which primarily affects the low vapor range of 0.05 to about 0.5%. The corrected biases were up to 40% of the WMO inter-laboratory compatibility goals (0.037 ppm for CO2 and 0.85 ppb for CH4). This definitely contributes to the community efforts to meet the WMO inter-laboratory compatibility goals for accurate GHG measurements. As the biases discussed here are small, many factors could have an impact on the significance of the results. In my opinion, the authors did a good job in bringing up and discussing the potential issues, however, failed to present them in a

clearly and well-structured way. Therefore, I can recommend publication of the paper in AMT after addressing the following concerns.

General The results of the enhanced water correction function for CO2 are not convincing. It will be helpful to summarize and list in a table all the factors that may cause a bias on the order of 0.037 ppm for CO2 and 0.85 ppb for CH4, and provide reasonable estimates of their associated uncertainties. For example, factors that may affect CO2 on this order of magnitude include

1) tank regulator effects that cause CO2 coming out of tanks drifting 2) uncertainties introduced by the sensitivity of CO2 to cavity pressure, e.g. 0.502 ppm/Torr (Table 2) was derived for Picarro #3, and 0.466 ppm/Torr was reported by Filges et al., 2015; 3) solubility of CO2 in water; 4) adsorption of CO2 by magnesium perchlorate, especially under changing pressure. In addition, the specified standard errors in the existing tables provide little information, as they are derived from the fit assuming statistical noise only, and are usually much lower than the overall uncertainties associated with the numbers.

Detailed comments:

Page 3 Line 9 The considerable amount of water used (500 ml) here will affect CO2 mole fractions. Has this effect been characterized?

Page 4 Line 22 Was there no offset in the cavity pressure compared to the external pressure measurement?

Page 9 Section 3.5.2 Which model was used? III or IV?

Page 13 Line 17 – 18 It is not clear what is said here. Rephrase the sentence.

---

## Author Comment (AC2) · 22 Sep 2017

Friedemann Reum, Christoph Gerbig, Jost V. Lavric, Chris W. Rella and Mathias Göckede

**Summary statement**

The reviewer highlighted the need to include a more thorough estimation of uncertainties in our measurements. We present this information here alongside responses to the other comments of the review.

**Reviewer's comment:**

General

The results of the enhanced water correction function for CO2 are not convincing. It will be helpful to summarize and list in a table all the factors that may cause a bias on the order of 0.037 ppm for CO2 and 0.85 ppb for CH4, and provide reasonable estimates of their associated uncertainties. For example, factors that may affect CO2 on this order of magnitude include

1) tank regulator effects that cause CO2 coming out of tanks drifting 2) uncertainties introduced by the sensitivity of CO2 to cavity pressure, e.g. 0.502 ppm/Torr (Table 2) was derived for Picarro #3, and 0.466 ppm/Torr was reported by Filges et al., 2015; 3) solubility of CO2 in water; 4) adsorption of CO2 by magnesium perchlorate, especially under changing pressure.

**Author's response:**

As the reviewer points out, our results were in part inconsistent especially for $CO_2$. In the manuscript, we discussed this issue extensively and summarized our considerations in the abstract: "The magnitude of the effect varied across instruments and appeared to be negligible for some, and our experimental results were more robust for $CH_4$ than for $CO_2$. Thus, correction coefficients should be determined for each analyzer individually."

We thank the reviewer for the suggestions on possible causes of biases on the same order as the observed effect. We have considered such biases as reasons for the inconsistent $CO_2$ results, since they may have overshadowed the pressure effect presented here, resulting in the failure to detect said effect on $CO_2$ in two instruments. We discussed this subject only briefly in Sect. 4.3 and 4.4 of the manuscript. However, as pointed out therein, such a bias would either have systematically counteracted the pressure effect (which is unlikely), or the pressure effect of these instruments was smaller than for the other (in line with the statement on this issue cited above).

In the following, we provide error estimates for the effects suggested by the reviewer.

1) Tank regulator effects that cause $CO_2$ coming out of tanks drifting

We avoided probing water levels in only strictly ascending or descending order to minimize the effect of drifts of $CO_2$ and $CH_4$ on our results. We assess whether such drifts were present in the following paragraph.

Carbon dioxide and $CH_4$ measurements from three experiments (Picarros #3, #4 and #5) have been analyzed in the manuscript (the experiment with Picarros #1 and #2 were intended for characterizing the pressure effect only, and the trace gas data from these experiments were not suitable for analysis). Picarro #3 displayed the pressure effect on $CH_4$, #5 displayed the pressure effect on both $CO_2$ and $CH_4$, while #4 displayed neither. For Picarro #3, dry air measurements were obtained between wet air measurements, which can be used to assess drifts. During the experiments with Picarros #4 and #5, only one dry air measurement has been obtained. Hence, we use the temporal variations of residuals to the water correction fit to assess drifts for these two experiments here.

The summary of the drifts over the course of the experiments is provided in Table 1. No statistically significant drift of $CO_2$ has been observed. Only the drift of $CH_4$ during the experiment with Picarro #5 was statistically significant ($p<0.05$). The drift has no considerable effect on the correction of the pressure effect, because the low water vapor levels relevant for the effect were all probed during one short period (e.g. Fig. 2 in author's response to Reviewer Comment 1 from July 7, 2017).

Table 1: $CO_2$ and $CH_4$ drifts during the water correction experiments. Picarro #3: Dry air measurements. Picarros #4 and #5: Water-corrected wet air measurements. The drifts are expressed as difference between the first and last observation based on a linear fit.

| Picarro | CO2 drift [ppm] | CO2 drift p-value | CH4 drift [ppb] | CH4 drift p-value |
|---------|-----------------|-------------------|-----------------|-------------------|
| #3 | -0.014 | 0.08 | -0.18 | 0.09 |
| #4 | 0.010 | 0.54 | -0.36 | 0.08 |
| #5 | 0.003 | 0.79 | -0.47 | 1.5e-5 |

2) Uncertainties introduced by the sensitivity of CO2 to cavity pressure

We agree that the total uncertainties of the relationships between external pressure measurement and $CO_2$ and $CH_4$ mole fractions are larger than the standard errors reported in Table 2 of the manuscript (see also our response to the next comment of the reviewer, below). We performed two such calibrations for Picarro #3, one with dry air and a second one with wet air. The results differed by a few percent (manuscript Sect. 3.1):

($CO_2$: +5 %, $CH_4$: -2 %, cavity pressure: +1 %). We concluded from these numbers that no systematic differences between dry and wet air are present in these relationships (see manuscript). The effect of these uncertainties on observed mole fractions is negligible, since the relative differences translate to relative differences of the absolute size of the pressure effect (i.e. 0.037 ppm $CO_2$ and 0.85 ppb $CH_4$ in our experiments). Hence, they are on the order of 0.002 ppm $CO_2$ and 0.02 ppb $CH_4$ for ambient mole fractions. These uncertainties, and the uncertainty of the pressure scale were not observable in our experiments and too small to explain the differences in the effects observed between instruments.

We will add this information to the revised manuscript by making the following change to Sect. 3.1:

Original (page 4 lines 20-21):

"The slopes in wet air (3 % $H_2O$) were measured for Picarro #3 and were very similar to the slopes in dry air ($CO_2$: +5 %, $CH_4$: -2 %, cavity pressure: +1 %)."

Changed to:

"The slopes in wet air (3 % $H_2O$) were measured for Picarro #3 and were very similar to the slopes in dry air ($CO_2$: +5 %, $CH_4$: -2 %, cavity pressure: +1 %). The effect of these uncertainties on observed mole fractions is negligible, since the relative differences translate to relative differences of the absolute size of the pressure effect (i.e. 0.037 ppm $CO_2$ and 0.85 ppb $CH_4$ in our experiments; see Sect. 3.3 and 3.5.1). Hence, the uncertainties are on the order of 0.002 ppm $CO_2$ and 0.02 ppb $CH_4$ for ambient mole fractions. These uncertainties, and the uncertainty of the pressure scale were too small to be observable in our experiments."

The reviewer remarked that the sensitivity of $CO_2$ to cavity pressure changes we found was different from the one in Filges et al. 2015. This is not the case, since the difference is due to the fact that we reported the sensitivities with respect to different mole fractions than Filges et al., with respect to the external pressure measurement instead of cavity pressure, and for another instrument. However, a direct comparison can be made since in fact Picarro #1 is the instrument used by Filges et al. To avoid confusion, we switch to reporting the sensitivities as fractional changes as in Kwok et al. 2015. In Table 2, we report the sensitivities of the instruments in our study, the findings of Filges et al. 2015, and the results of Kwok et al. 2015. Our results for Picarro #1 are very well compatible with those by Filges et al. 2015. The mostly small differences between the sensitivities obtained for the different instruments may be due to differences in the absolute pressure calibration of the internal pressure sensors of the analyzers. Only the sensitivity of $CH_4$ from Kwok et al. differs strongly from the other results. This is likely due to uncertainties of the experiment by Kwok et al. (C. W. Rella, personal communication). We did not explore the differences further in our study.

We will update the way we reported the sensitivities of $CO_2$ and $CH_4$ to pressure in Table 2 of the manuscript with the information presented here.

Table 2: Sensitivities of $CO_2$ and $CH_4$ to changes of internal cavity pressure.

| Picarro | $\frac{\Delta CO_{2,frac}}{\Delta P}$ [Torr$^{-1}$] | $\frac{\Delta CH_{4,frac}}{\Delta P}$ [Torr$^{-1}$] |
|---|---|---|
| #1 | $(1.20 \pm 0.02) \times 10^{-3}$ | $(4.26 \pm 0.06) \times 10^{-3}$ |
| #1 (Filges et al. 2015) | $1.20 \times 10^{-3}$ | $4.3 \times 10^{-3}$ |
| #2 | $(1.258 \pm 0.002) \times 10^{-3}$ | $(4.372 \pm 0.002) \times 10^{-3}$ |
| #3 | $(1.200 \pm 0.002) \times 10^{-3}$ | $(4.326 \pm 0.005) \times 10^{-3}$ |
| (Kwok et al. 2015) | $1.3 \times 10^{-3}$ | $3.5 \times 10^{-3}$ |

3) Solubility of CO2 and CH4 in water / drifts

As explained in the manuscript and further elaborated on in our response to Reviewer Comment 1 (July 7, 2017), the strategies to avoid effects of dissolution in or outgassing from the water reservoir were to wait for stable signals and vary the order in which water vapor levels were probed. We acknowledged the possibility that equilibration processes might have affected our results and gone unnoticed if they were on timescales much longer than one hour. In the following paragraph we look for such equilibration processes by analyzing drifts in the available data. In particular, we analyzed drifts of $CO_2$, $CH_4$, $H_2O$ and pressure during the intervals that were used for further analysis and looked for correlations with residuals of the water correction fit. If a drift that affected the mole fractions were detectable, it would have a negative correlation with the fit residuals, since a downward (upward) trend would mean the recorded mole fraction was too high (too low). As can be seen from Fig. 2, no such trend could be detected. However, we did find correlations between $CO_2$, $CH_4$ and external pressure measurement drifts (Fig. 3). Hence, these correlations are due to drifts of cavity pressure, meaning that perfect stability was not reached. Despite this relationship, we could not find evidence that these drifts affected our results in a systematic way, as they were random with mean effects near zero and they did not explain the residuals of the water correction fit (Fig. 1 and Fig. 2). Furthermore, we investigated the drifts during the experiment with Picarro #3 more closely. The drifts depended on the data selection used for analysis; the last 10, 15 and 20 minutes of each probing interval were tested (Fig. 4). This indicates that these drifts were not part of a longer-term trend, but short-term variations. This is supported by the similarity of the pressure effect of the three experiments despite these different drifts (manuscript Sect. 3.2). Nonetheless, we will stress in a revised manuscript that there may be effects on longer timescales than those probed here, and encourage future research on this issue:

Original (page 12 lines 1-3):

"If this explanation were true, the systematic difference between dry air and wet air trace gas mole fractions would have precisely compensated for the pressure bend, which seems unlikely."

Changed to:

"If this explanation were true, the systematic difference between dry air and wet air trace gas mole fractions would have precisely compensated for the pressure bend, which seems unlikely. Nevertheless, to exclude that such equilibration effects influence the water correction, we encourage future research on this topic."

To estimate the contributions of different error sources to the overall error of our $CO_2$ and $CH_4$ measurements, we use the relationships between mole fraction drifts and pressure drifts as a measure for uncertainties associated with cavity pressure equilibration. Furthermore, we assume that the residual drift after subtracting the pressure equilibration drift was associated with other effects such as equilibration with the water reservoir,

potentially due to temperature instability. The uncertainties obtained in this way are summarized in Table 3.

[Figure]

**Fig. 1: Drifts of data during the interval used for analysis (last 15 minutes of each probing interval) vs H2O (Picarro #3).**

[Figure]

**Fig. 2: Water correction fit residuals vs drifts (data as in Fig. 1)**

[Figure]

Fig. 3: CO₂ and CH₄ drifts vs pressure drift (data as in Fig. 1)

[Figure]

[Figure]

**Fig. 4: Pressure drifts vs H₂O. From top left to bottom: Picarro #1, #2, #3 (10 min), #3 (15 min), #3 (20 min). The times refer to the amount of data at the end of each probing interval used.**

**Table 3: Error estimates based on drifts of trace gas and pressure in the data used for analysis (last 15 minutes of the probing intervals). The total errors are very similar to the estimates based on fit residuals (manuscript Sect. 3.3), confirming their validity.**

|  | $CO_2$ [ppm] | $CH_4$ [ppb] |
|---|---|---|
| Pressure equilibration | 0.012 | 0.16 |
| Other, incl. solubility | 0.009 | 0.13 |
| Sum | 0.015 | 0.21 |
| Comparison: fit residuals | 0.014 | 0.17 |

4) Adsorption of CO2 by magnesium perchlorate, especially under changing pressure.

As explained in our response to Reviewer Comment 1 (July 7, 2017), the magnesium perchlorate cartridge was never installed upstream of the Picarro analyzer during the experiments from which we analyzed the $CO_2$ and $CH_4$ mole fraction data. Therefore, there is no error associated with adsorption of $CO_2$ by magnesium perchlorate in our analyses.

**Reviewer's comment:**

In addition, the specified standard errors in the existing tables provide little information, as they are derived from the fit assuming statistical noise only, and are usually much lower than the overall uncertainties associated with the numbers.

**Author's response:**

We agree that the standard errors of the fit parameters reported in Tables 2-5 of the manuscript may underestimate the uncertainties of the quantities presented. This is the reason why these uncertainty estimates were not used for the key quantities of the paper, and thus have little significance in the first place. In our opinion, the best estimate of the uncertainties of an instrument calibration such as the water correction presented here is to repeatedly perform calibration experiments and use the differences between individual calibrations to derive the uncertainties associated with the calibration. This approach is necessary to quantify uncertainties that are not detectable from a single calibration

experiment, such as variations with time and environmental conditions. Thus, the full uncertainty estimate of the water correction must be assessed for instruments individually, while the scope of this paper is to provide a method to correct one specific effect.

In our manuscript, we recommended one parameter, the pressure bend position $h_p$, for use with water correction functions of other analyzers in the absence of sufficient data to constrain it. In line with our statement about uncertainties above, we estimated the uncertainty of $h_p$ as the standard deviation of the estimates for the three individual instruments, reflecting the variability of this quantity. The error associated with this parameter and its usage for other instruments was extensively discussed in Sect. 3.5.1. Therefore, we do not think it is necessary to change the uncertainties reported in Tables 2-5 of the manuscript.

**Reviewer's comment:**

Detailed comments:

Page 3 Line 9 The considerable amount of water used (500 ml) here will affect CO2 mole fractions. Has this effect been characterized?

**Author's response:**

In the manuscript we failed to clarify which amount of water was used for which experiment. During experiments from which $CO_2$ and $CH_4$ mole fractions were analyzed and presented, the amount of water used was considerably smaller than 500 ml. In the experiment with Picarro #3, 15 ml were used, with Picarros #4 and #5 the amount was about 40 ml. The 500 ml have been used in experiments from which only pressure data was interpreted (Picarros #1 and #2). We will clarify this by making the following change in Sect. 2.1 of the manuscript:

Original (page 3 line 8-9):

"Air in the other line was directed through a gas washing bottle (glass) containing deionized water (depending on bottle size, about 15 ml to about 500 ml were used in the experiments presented here)."

Changed to:

"Air in the other line was directed through a gas washing bottle (glass) containing deionized water. The amount of water used varied. In the experiments from which only pressure data were interpreted (Picarros #1 and #2), 500 ml of deionized water were used. In the experiments from which trace gas data were interpreted, the amount of water was reduced to ensure faster equilibration of the water reservoir with the air stream. In the experiment with Picarro #3, 15 ml were used, and in the experiments with Picarros #4 and #5 the amount was 40 ml."

As the reviewer pointed out, streaming air through liquid water alters the mole fractions of the gas stream by dissolution or outgassing until equilibrium is reached. The equilibrium shifts with temperature and pressure. In his/her comment from July 6, 2017, Anonymous Referee #1 raised questions about temperature and pressure stability during our experiments as well. We addressed these questions together with equilibration of the gas

stream with the water reservoir in the gas washing bottles and the effects they might have had on our findings on pages 5-10 of our response to Reviewer Comment 1 (July 7, 2017). We conclude that temperature and pressure stability were sufficient and did not affect our findings.

**Reviewer's comment:**

Page 4 Line 22 Was there no offset in the cavity pressure compared to the external pressure measurement?

**Author's response:**

Yes, as mentioned in Sect. 2.1 (page 3, line 26-27), there was an offset between cavity pressure and external pressure measurement. The transformation of external pressure readings to cavity pressure was explained in Sect. 3.2 of the manuscript. To clarify the procedure, we will make the following change in this section:

Original (page 4 line 26 – page 5 line 3):

"To calculate a "corrected cavity pressure" from the external pressure measurement, pressure readings for dry air before and after each wet air measurement were interpolated to the times of the wet air measurements. The deviations between the wet air pressure values and the interpolated dry air pressure values were multiplied with the slope described in Sect. 3.1, and added to the dry air cavity pressure of 140 Torr."

Changed to:

"As explained in Sect. 3.1, the external pressure measurement was modeled as a linear function (slope and offset) of internal cavity pressure. Therefore, to calculate a "corrected cavity pressure" from the external pressure measurement, pressure readings for dry air before and after each wet air measurement were first interpolated to the times of the wet air measurements. Then the deviations between the wet air pressure values and the interpolated dry air pressure values were multiplied with the slope, and added to the dry air cavity pressure of 140 Torr."

**Reviewer's comment:**

Page 9 Section 3.5.2 Which model was used? III or IV?

**Author's response:**

The water correction used was that described in Sect. 3.1. It was model (iv) in that a fixed value for the pressure bend position $h_p$ was used. To clarify this, we will make the following changes:

Original (page 7 line 29-30):

"Therefore, we used the mean $h_p$ from the three experiments with external pressure monitoring, Eq. (4), and investigated the uncertainty associated with this procedure."

Changed to:

"Therefore, we used model (iv) with $h_p$ fixed to its mean value from the three experiments with external pressure monitoring, Eq. (4), and investigated the uncertainty associated with this procedure."

Original (page 9 line 4-5):

"In this section, we describe the impact of the improved water correction on hourly averages of $CO_2$ and $CH_4$ data from Ambarchik over the years 2015 and 2016"

Changed to:

"In this section, we describe the impact of the improved water correction, which was derived in Sect. 3.5.1, on hourly averages of $CO_2$ and $CH_4$ data from Ambarchik over the years 2015 and 2016"

**Reviewer's comment:**

Page 13 Line 17 – 18 It is not clear what is said here. Rephrase the sentence.

**Author's response:**

We will make the following change to clarify this sentence:

Original (Page 13 Line 17 – 18):

"The results indicated small impacts of observation biases considerably larger than the WMO goals on annual flux budgets at continental scales. "

Changed to:

"In some cases, the observation biases assessed in these studies were considerably larger than the bias corrected by using the improved water correction presented here. The studies indicated small impacts even of these comparatively larger observation biases on annual flux budgets at continental scales. "